# Statistical injury prediction for professional sumo wrestlers: Modeling and perspectives

**Shuhei Ota**[1]*, **Mitsuhiro Kimura**[2]

**1** Department of Industrial Engineering and Management, Kanagawa University, Yokohama, Kanagawa, Japan, **2** Department of Industrial and Systems Engineering, Hosei University, Faculty of Science & Engineering, Tokyo, Japan

* ota@kanagawa-u.ac.jp

**Data Availability Statement:** All data are available from the Sumo reference's database (URL: http://sumodb.sumogames.de).

**Funding:** This study was supported by JSPS KAKENHI Grant Numbers 19K04892 and

## Abstract

In sumo wrestling, a traditional sport in Japan, many wrestlers suffer from injuries through bouts. In 2019, an average of 5.2 out of 42 wrestlers in the top division of professional sumo wrestling were absent in each grand sumo tournament due to injury. As the number of injury occurrences increases, professional sumo wrestling becomes less interesting for sumo fans, requiring systems to prevent future occurrences. Statistical injury prediction is a useful way to communicate the risk of injuries for wrestlers and their coaches. However, the existing statistical methods of injury prediction are not always accurate because they do not consider the long-term effects of injuries. Here, we propose a statistical model of injury occurrences for sumo wrestlers. The proposed model provides the estimated probability of the next potential injury occurrence for a wrestler. In addition, it can support making a risk-based injury prevention scenario for wrestlers. While a previous study modeled injury occurrences by using the Poisson process, we model it by using the Hawkes process to consider the long-term effect of injuries. The proposed model can also be applied to injury prediction for athletes of other sports.

## Introduction

Sumo, a traditional sport dating back about 1500 years in Japan, is a form of competitive full-contact wrestling where two wrestlers fight in a ring [1, 2]. Since 1958, six grand tournaments of professional sumo wrestling that consists of six divisions have been held per year. Each tournament runs for 15 days, and each wrestler in the top two divisions has one match per day while the lower-ranked wrestlers compete in seven bouts about once every two days [3]. There were over 600 wrestlers from the bottom to top divisions in 2022, and the capacities of the top and second from top divisions were 42 and 28, respectively. Regarding worldwide interest in sumo, today, there are national clubs of amateur sumo wrestling in over 84 countries [4]. Member countries of the International Sumo Federation host an international World Sumo Championship each year. For various interesting papers on sumo, see e.g., [3, 5–7].

Injuries commonly occur in most professional sumo wrestlers and negatively affect their performance. The average weight of wrestlers is over 160 kg, so the physical load yielded in a

21K14373 (Funder's website: https://www.jsps.go.jp/english). The grants' representatives were MK and SO, respectively. The funder had no role in study design, data collection and analysis, decision to publish, or preparation of the manuscript.

**Competing interests:** The authors have declared that no competing interests exist.

bout frequently causes severe injury [8]. In particular, an anterior cruciate ligament (ACL) injury [9] is a common traumatic injury in professional wrestlers [7]. Wrestlers have a higher risk of second injuries after ACL reconstruction than athletes of other sports [7]. Consequently, wrestlers are sometimes required to be absent from grand sumo tournaments to heal severe injuries.

*Kyujo* is a state in which a sumo wrestler is absent from a bout in a grand sumo tournament due to injury or illness [2]. A *kyujo* occurrence is officially recorded for each wrestler, bout and tournament [2]. For example, in 2019, an average of 5.2 out of 42 sumo wrestlers in the top division were *kyujo* in each grand sumo tournament due to injury [10]. Because the *kyujo* for a bout is regarded as a loss, wrestlers tend not to be *kyujo* to maintain their rank score even if they have injuries in general. Consequently, wrestlers tend to have a serious injury when they are *kyujo*. In addition, as the number of *kyujo* occurrences of wrestlers increases, professional sumo wrestling becomes less interesting for sumo fans, requiring systems to prevent future occurrences. To prevent injuries, wrestlers and their coaches need to be better informed about the risk of injuries; however, such an indicator is unavailable.

Injury prediction [11, 12], which is used to predict potential injuries in the future, is a useful way to communicate the risk of injuries for athletes. It has been a challenging research topic in sports analytics owing to the practical use of big data such as activities and health conditions of athletes [13, 14]. Forecasting/identifying athletes who are prone to injury is useful and practical from both financial and health aspects, which is why the use of artificial intelligence (AI) knowledge has become increasingly popular in studies on sport-related injuries. Therefore, many existing studies developed injury prediction methods using risk factors that can be divided into modifiable and non-modifiable factors [9, 11]. Non-modifiable factors are those that cannot be altered by any means, such as age, gender, and previous injury history [15]. Modifiable factors are those that are potentially modifiable through physical training or behavioral approaches, such as body mass, muscle strength, and flexibility.

Related works of injury prediction have mainly used machine learning with risk factors. Hulin et al. [16] found that the acute:chronic workload ratio predicts injuries in elite rugby league players. Gabbett [17] modeled relationships between the training load and likelihood of injury to predict injuries in elite collision sport athletes. Rossi et al. [18] proposed an injury prediction method for professional soccer players by using GPS data and a machine learning technique. Rommers et al. [19] presented a machine learning model to predict injuries in elite-level youth football players with reasonable accuracy. However, injury prediction based on machine learning is not applicable for sumo wrestlers at the current stage because there is insufficient data except for *kyujo* records to build machine learning models in sumo wrestling.

Historical data of injury occurrences still provide important information for injury prediction [20, 21]. Such data can be modeled by stochastic processes (e.g., Poisson process), which enable us to simulate how injuries occur as a consequence of fatigue, previous trauma, lack of physical preparation, and bad luck. For example, [21] developed a Poisson process model of injury prediction for schoolboy rugby. The model provides the average probability of an injury occurrence. However, it is only applicable under the condition that all injuries occur independently because of its memoryless property [21, 22]. In other words, it cannot consider the long-term effects of injuries. Therefore, the Poisson process model is unsuitable for injury prediction in sumo wrestlers who are prone to re-injury due to previous injuries. For more accurate injury prediction, we should consider the long-term effect of injuries in the statistical model.

Thus, we propose a new statistical model of injury prediction for professional sumo wrestlers by considering the long-term effect of injuries. Recurrent events like injury occurrences are commonly observed in clinical studies [23, 24]. While previous research represented injury occurrences by using the Poisson process model, we describe it by using the Hawkes process

model [22, 25]. The Hawkes process is a statistical model that is frequently used in risk analysis of finance [26, 27], epidemiology [28, 29], and seismology [30, 31]. It can express that past events can increase the likelihood of future events occurring. For example, in seismology, the Hawkes process well predicts times of the subsequent earthquake occurrences by considering the effect of aftershocks. In the same way, we use the Hawkes process to consider the long-term effect of injuries.

By using the proposed model, we perform injury prediction for sumo wrestlers by predicting potential *kyujo* occurrences in the future. The proposed model provides (i) the estimated probability of a *kyujo* occurrence for each wrestler and (ii) the predicted number of *kyujo* wrestlers due to injury in a grand sumo tournament. In addition, the estimated probability of a *kyujo* occurrence can be used to make a risk-based injury prevention scenario for wrestlers. Such a means of risk-based scenario planning is unique and beneficial for wrestlers and their coaches. Moreover, because the proposed model uses only time and injury history data, one can apply the proposed model to other sports.

The novelty of this paper is that this is the first application of the Hawkes process to an injury prediction model. The proposed model using the Hawkes process can consider the long-term effect of injuries while previous models using the Poisson process model cannot. Therefore, the proposed model is suitable for injury prediction of athletes who are frequently injured like professional sumo wrestlers.

## Materials and methods

In this section, we first illustrate the characteristic properties of *kyujo* occurrences of professional sumo wrestlers. We then construct an injury prediction model for wrestlers based on the features. For the purpose of this study, we define an injury as a cause of *kyujo*. Thus, in injury prediction, we ignore relatively minor injuries that do not make wrestlers *kyujo*.

### Injury data collection

To construct an injury prediction model for sumo wrestlers, we analyze *kyujo* occurrences. For this analysis, we refer to an open database [10] that consists of player hours at *kyujo* occurrences for each wrestler where player hours (unit: 1,000 bouts) are the sum of the number of wins, losses, and *kyujo* occurrences at a given period. We extracted two data sets from this open database as Data-A and Data-B.

Data-A, which includes a total of $n = 209$ sumo wrestlers who played their first match between 1973 and 2003 and belonged to the top divisions more than once, is used for estimating the parameters of the proposed model. Note that Data-A does not include data about wrestlers who had been dismissed once and reinstated afterward. Data-B, which includes all wrestlers who belonged to the top division in the grand sumo tournament of November 2020, is used for an illustrative example of the proposed injury prediction. To simplify the analysis in the next section, we regard consecutive *kyujo* occurrences as one *kyujo* occurrence.

The *kyujo* data are partially subjected to informative censoring [32] which is also called dependent censoring [33]. That is, while sumo wrestlers likely end their careers for various reasons, the injuries leading to *kyujo*, repeated *kyujo*, or long player hours are likely the proximal cause in many cases. This type of censoring may have a negative effect on the results of the injury prediction. However, it is reasonable that with multiple events per player, the effects might be minimal. The histogram of the number of *kyujo* occurrences is shown in Fig 1, where 88% of wrestlers had one or more *kyujo* occurrences until they retired. In addition, a retirement and *kyujo* can be considered the same because wrestlers should be absent from a bout in both situations. Therefore, we assume that a retirement can be regarded as a *kyujo* occurrence.

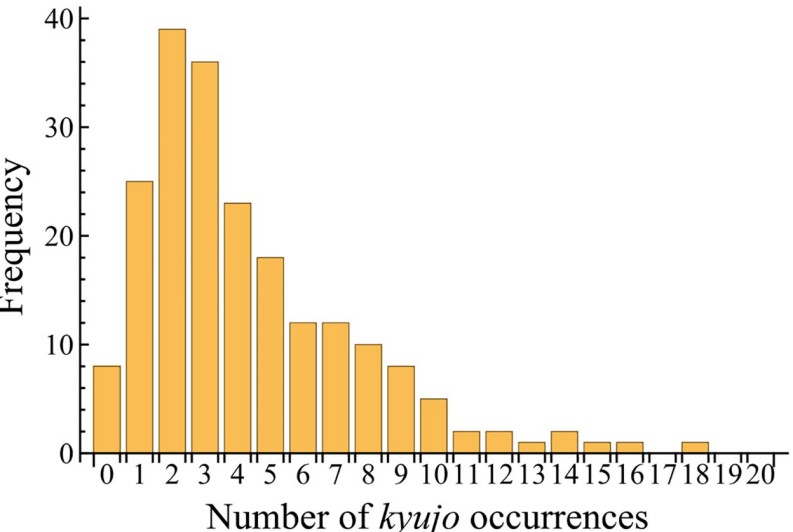

**Fig 1. Histogram of the number of *kyujo* occurrences.**

We will discuss that this assumption does not affect the performance of the injury prediction in the subsection on model validation.

## Feature extraction

As useful information for modeling, we present the characteristic properties of *kyujo* occurrences of professional sumo wrestlers.

Fig 2 illustrates *kyujo* occurrences for player hours of each sumo wrestler in Data-A. In this figure, we can observe a feature that many *kyujo* occurrences happen sequentially. This fact indicates that wrestlers are likely to be *kyujo* again after one due to long-term effects of past injuries. In fact, occurrences of ipsilateral reinjury and contralateral ACL injury after ACL reconstruction in professional sumo wrestlers are relatively higher than those reported in previous studies on athletes in other sports [7].

Another feature of *kyujo* occurrences of sumo wrestlers can be found through the failure rate (see [34, 35] for details of the failure rate). For given player hours $t \, (> 0)$, we define the failure rate $r(t)$ for a *kyujo* occurrence as follows.

$$r(t) = \frac{\# \, \{\text{sumo wrestlers who play at } t \text{ and are } kyujo \text{ on } (t, t + dt]\}}{\# \, \{\text{sumo wrestlers who play at } t\}}, \tag{1}$$

where $dt > 0$. Thus, the failure rate $r(t)$ express the probability that wrestles becomes *kyujo* in the interval $t$ to $t + dt$. Fig 3 represents the behavior of $r(t)$ for the sumo wrestlers. In this observation, the failure rate has two distinct stages, namely Stages I and II. In Stage I, the failure rate is at a low and stable level until the average player hours of the first *kyujo* occurrence (i.e., for $[0, \tau = 0.342)$). After $\tau$, the failure rate increases as $t$ increases in Stage II. The features of the failure rate in Stages I and II are known as the random failure period and wear-out failure period of products in reliability engineering [36], respectively.

Thus, the properties of *kyujo* occurrences are summarized as (i) a *kyujo* tends to occur successively, (ii) the failure rate is stable until the average player hours of the first *kyujo* occurrences, and (iii) the failure rate increases as player hours increase after the first *kyujo*

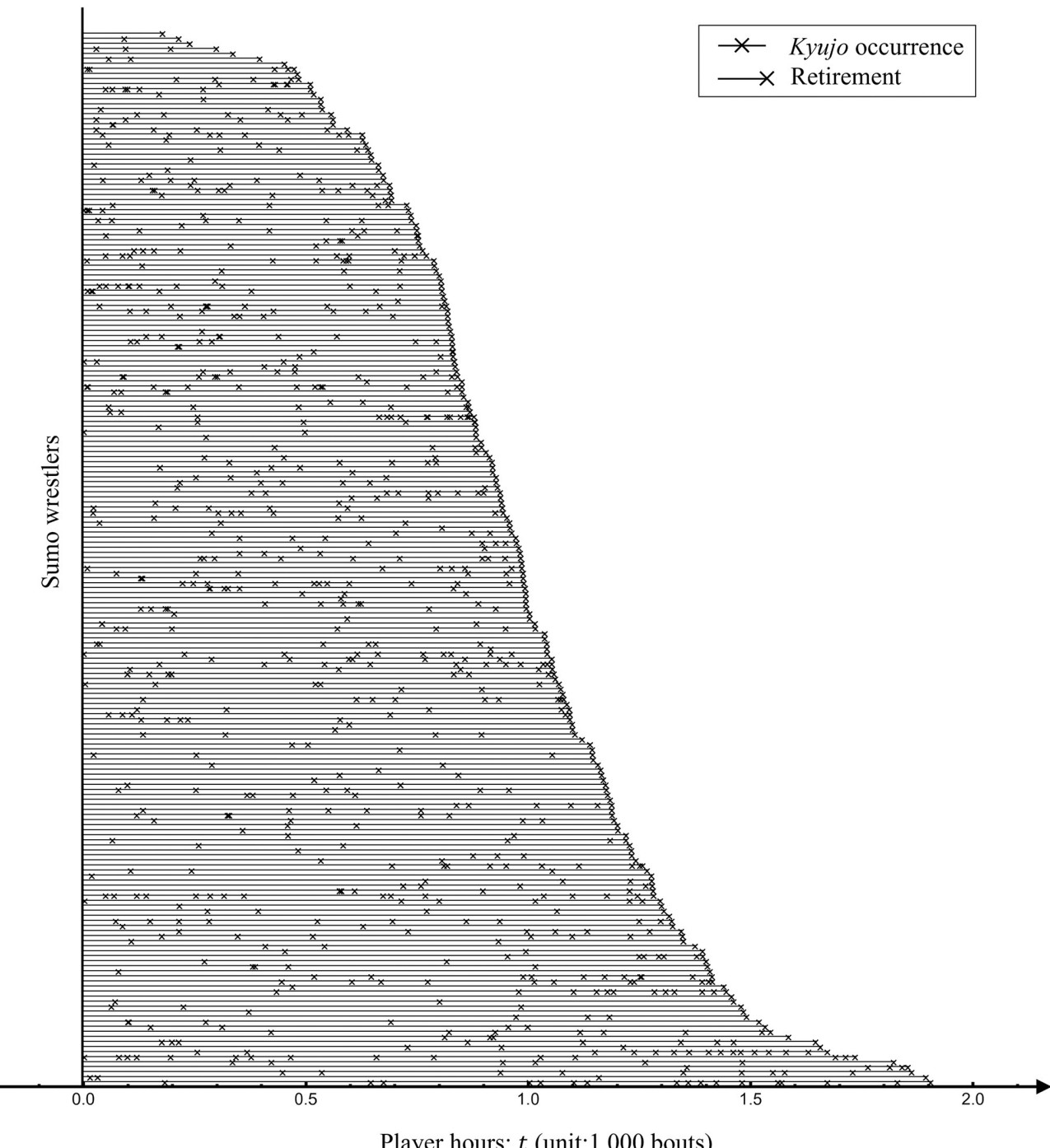

**Fig 2. Structure of *kyujo* occurrences of 209 professional sumo wrestlers in Data-A.** Each line represents data of a wrestler. $t = 0$ means the first bout of each wrestler. Wrestlers are sorted in ascending order of player hours at their retirement.

occurrence. These properties must be considered in injury prediction models to ensure accurate prediction.

## Statistical modeling

In this section, we first briefly introduce the Poisson process model. We then propose a statistical model of *kyujo* occurrences for sumo wrestlers based on the features of the data we discussed. The models require player hours at the *kyujo* occurrences for each wrestler. Let $T_{ij}$ be the random variable that represents the player hours at the *j*-th *kyujo* occurrence of the *i*-th sumo wrestler for $i = 1, 2, \ldots, n$ and $j = 1, 2, \ldots$. Suppose $T_{ij}$'s are characterized by an intensity function [22] that is the instantaneous likelihood of a *kyujo* occurrence at player hours *t*. The intensity function is usually used to analyze failure event data (see e.g., [37, 38]).

The models also require the following assumptions.

**Assumption 1**: All *kyujo* occurrences of sumo wrestlers follow an independent and identical stochastic process.

**Assumption 2**: *Kyujo* periods are shorter than play periods and can be ignored.

If $T_{ij}$ obeys a homogeneous Poisson process (HPP), the intensity function of the process, denoted by $\lambda(t)$, is determined by a constant value as $\lambda(t) = \lambda_0$ for $\lambda_0 > 0$. Namely, the HPP model assumes that the likelihoods of *kyujo* occurrences are homogeneous and independent of previous *kyujo* occurrences. The HPP model is simple and suitable to predict injury occurrences if $T_{ij}$'s are statistically independent for $j = 1, 2, \ldots$. However, this model cannot consider the heterogeneous behavior of the intensity function as the failure rate in Stage II because of the memoryless property of the Poisson process.

Now, we propose an injury prediction model using a Hawkes process [22, 25, 39] to capture the behavior of the failure rate in Stages I and II. The proposed model assumes that $T_{ij}$ follows a Hawkes process determined by the following intensity function $\lambda(t|H_t)$.

$$\lambda(t|H_t) = \begin{cases} \lambda_0 & (t \leq T_{i1}) \\ \lambda_0 + ab(t - T_{i1})^{b-1} + \sum_{T_{ij} < t} \alpha e^{-\beta(t - T_{ij})} & (t > T_{i1}) \end{cases}, \quad (2)$$

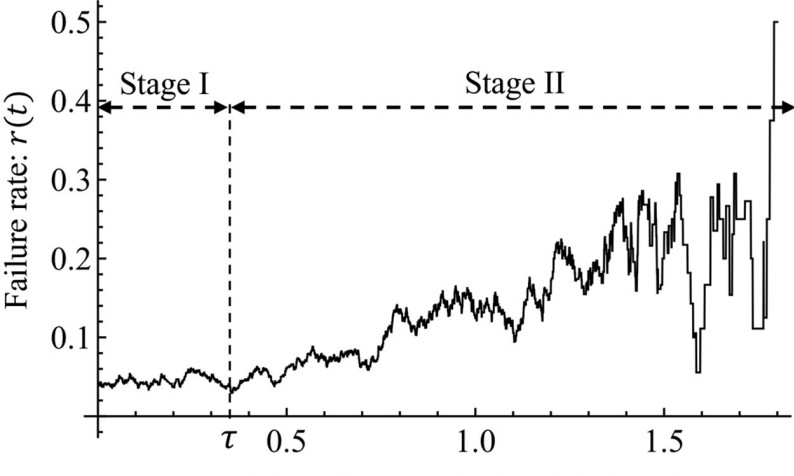

**Fig 3. Behavior of the failure rate *r(t)* for *dt* = 0.05, where *τ* = 0.342 is the average player hours of the first *kyujo* occurrence in Data-A.**

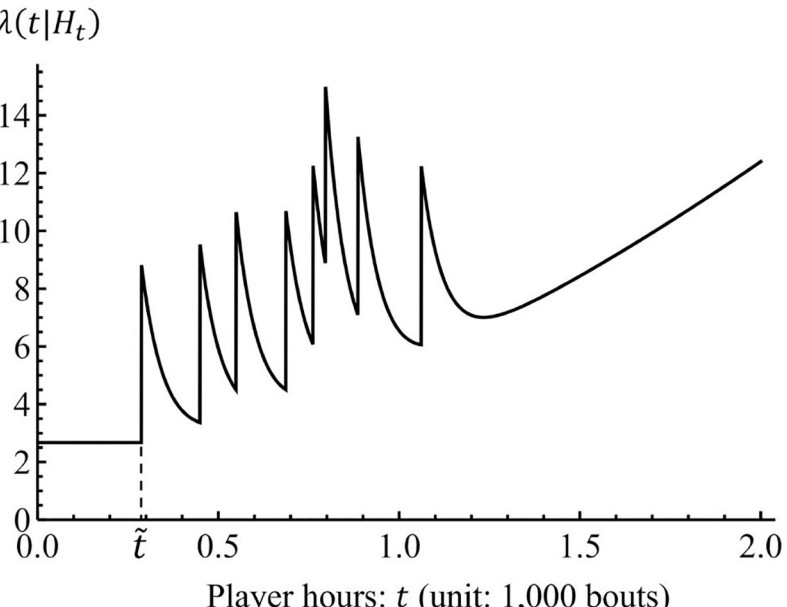

**Fig 4. Illustration of the proposed model ($b > 1$).**

where $i = 1, 2, \ldots, n, j = 1, 2, \ldots, \lambda_0, a, b, \alpha, \beta > 0$, and $H_t = \{T_{ij} | T_{ij} < t\}$ is the history of the point process on $[0, t)$, i.e., the set of all *kyujo* occurrences on $[0, t)$. This assumption represents that the likelihoods of *kyujo* occurrences are dependent on all previous *kyujo* occurrences for each sumo wrestler. Eq (2) is divided by cases to describe the behavior of the failure rate in Stages I and II. For $t \leq T_{i1}$, Eq (2) represents the stable behavior of the failure rate in Stage I, and the expected value of $T_{i1}$ is given by $1/\lambda_0$. As for $t > T_{i1}$, Eq (2) describes the increasing behavior of the failure rate in Stage II. Specifically, its second factor means that wrestlers are more likely to be injured as player hours increase, and the third one expresses the positive effect of the long-term effect of all past *kyujo* occurrences on the current value of the intensity. The player hours $T_{i1}$ is the change point of the intensity [40]. The proposed model generalizes the Poisson process model [21] because Eq (2) corresponds to the intensity function of the Poisson process if $a = \alpha = 0$. The model parameters should be statistically estimated from actual data. We describe an estimation method for the parameters in the next subsection.

Fig 4 represents an example of the behavior of the proposed model for $b > 1$. The intensity is constant until the player hours of the first *kyujo* occurrence $\tilde{\tau}$ from $t = 0$ and jumps by a value of $\alpha$. After the jump at $t = \tilde{\tau}$, the intensity exponentially decays until the player hours of the next *kyujo* occurrence, and the jump and decay repeat again and again. If the *kyujo* event does not occur, the intensity polynomially increases as shown after $t = 1.2$. Hence, the proposed model captures the features of *kyujo* occurrences.

The probability of a *kyujo* occurrence of a sumo wrestler in the future can be calculated by Eq (2). Let $p_{ij}(w|t)$ be the probability of the $j$-th *kyujo* occurrence of the $i$-th wrestler within player hours $w (> 0)$ under the condition that his player hours are $t$. Then, $p_{ij}(w|t)$ is given by

$$p_{ij}(w|t) = \begin{cases} 1 - e^{-\lambda_0 w} & (j = 1) \\ 1 - e^{-\int_t^{t+w} \lambda(s|H_t)ds} & (j \geq 2) \end{cases}. \tag{3}$$

In Eq (3), the injury probability of a sumo wrestler does not depend on the player hours $t$ if he has never had a *kyujo*. Otherwise, the probability depends on $w$, $t$, and $H_t$.

The probability $p_{ij}(w|t)$ is a useful criterion for each sumo wrestler to understand his injury risk at given player hours. For example, $p_{ij}(0.015|t)$ expresses the probability of the next *kyujo* occurrence in a grand sumo tournament (N.B. the period of a tournament corresponds to 0.015 player hours). Hence, one can estimate probabilities that wrestlers will be injured in a tournament by using the data that can be obtained before the tournament starts.

By using $p_{ij}(w|t)$, we can also predict the number of *kyujo* sumo wrestlers due to injury in the top division of a grand sumo tournament to where 42 wrestlers belong. Such a number is helpful for their coaches to understand an overview of wrestlers' injury risks. Let $X$ be a random variable that represents the number of *kyujo* wrestlers and $q_x \equiv \Pr[X = x]$ be the probability mass function of $X$ for $x = 0, 1, 2, \ldots, 42$. Let $t_i$ be the player hours of the $i$-th wrestler at the first bout in a given tournament. Suppose that each wrestler will be *kyujo* with the probability of $p_{ij}(w|t_i)$. Then, $X$ follows a Poisson binomial distribution [41], and $q_x$ is given by

$$q_x = \sum_{S' \in S_x} \left\{ \prod_{i \in S'} p_{ij}(0.015|t_i) \prod_{i \in (I \setminus S')} (1 - p_{ij}(0.015|t_i)) \right\}, \tag{4}$$

where $I = \{1, 2, \ldots, 42\}$ and $S_x$ are all subsets of $x$ integers that can be selected from $I$. In addition, the expectation and variance of $X$ are respectively as follows.

$$E[X] = \sum_{i=1}^{42} p_{ij}(0.015|t_i), \quad V[X] = \sum_{i=1}^{42} (1 - p_{ij}(0.015|t_i)) p_{ij}(0.015|t_i).$$

We can summarize injury risks of sumo wrestlers in a division of a grand sumo tournament by $q_x$, $E[X]$, and $V[X]$.

## Method of maximum likelihood

In this subsection, we explain the method of maximum likelihood to estimate parameters of a Hawkes process model $\lambda(t|H_t)$, i.e., $\lambda_0$, $a$, $b$, $\alpha$, and $\beta$.

Let $v_i$ be the total number of *kyujo* occurrences of the $i$-th sumo wrestler until his retirement. Then, the log-likelihood function of a Hawkes process with the intensity $\lambda(t|H_t)$, denoted by $\log \mathcal{L}$, is given by

$$\log \mathcal{L}(\lambda_0, a, b, \alpha, \beta | H_t) = \sum_{i=1}^{n} \left\{ \sum_{j=1}^{v_i} \log \lambda(t_{ij}|H_t) - \int_0^{t_{iv_i}} \lambda(s|H_t) ds \right\}. \tag{5}$$

According to Ogata [39], we obtain the following recursive formula of Eq (5).

$$\log \mathcal{L}(\lambda_0, a, b, \alpha, \beta | H_t) = \sum_{i=1}^{n} \left\{ \sum_{j=1}^{v_i} \log \left( h(t_{ij}|H_t) + \alpha R(j) \right) \right.$$
$$\left. - \int_0^{t_{iv_i}} h(s|H_t) ds - \sum_{j=1}^{v_i} \frac{\alpha}{\beta} \left( 1 - e^{(-\beta(t_{iv_i} - t_{ij}))} \right) \right\}, \tag{6}$$

where

$$h(t|H_t) = \begin{cases} \lambda_0 & (t \le t_{i1}) \\ \lambda_0 + ab(t - t_{i1})^{b-1} & (t > t_{i1}) \end{cases},$$

$$R(j) = \begin{cases} 0 & (j = 1) \\ e^{-\beta(t_{ij} - t_{ij-1})}(1 + R(j-1)) & (j \ge 2) \end{cases}.$$

By maximizing Eq (6), one can obtain $\hat{\lambda}_0, \hat{a}, \hat{b}, \hat{\alpha}$, and $\hat{\beta}$ as the maximum likelihood estimates of model parameters $\lambda_0$, $a$, $b$, $\alpha$, and $\beta$.

We note that the censoring of *kyujo* occurrences can be ignored in this estimation process because we assumed that retirements are the same as *kyujo* occurrences in the subsection on injury data collection. As a remark, one may use the inverse probability of censoring weights (IPCW) method [32] or copula-based approaches [33] to handle censoring. One relatively simple way to assess estimation performance for these methods is to set up simulation studies that implement the informative censoring.

## Results and discussion

In this section, we first start with model validation by estimating the parameters of the proposed model with Data-A. We then present illustrative examples of injury prediction for professional sumo wrestlers with Data-B. Fig 5 visualizes the overall structure of the model validation and injury prediction processes. In this figure, boxes represent processes, and arrows represent the input and output of the corresponding processes.

### Parameter estimation

To model the injury occurrence of professional sumo wrestlers, we estimate the parameters of Eq (2) from Data-A by the method of maximum likelihood [39, 42]. The parameters can be estimated by maximizing the log-likelihood function given by Eq (6). Let $\hat{\lambda}_0, \hat{a}, \hat{b}, \hat{\alpha}$, and $\hat{\beta}$ be the estimates of $\lambda_0$, $a$, $b$, $\alpha$, and $\beta$, respectively. Let $\hat{\lambda}(t|H_t), \hat{p}_{ij}(w|t)$, and $\hat{q}_x$ be the estimates of $\lambda(t|H_t), p_{ij}(w|t)$, and $q_x$ in which parameters $\lambda_0$, $a$, $b$, $\alpha$, and $\beta$ are replaced by their estimates, respectively.

To select the best model, we use the Akaike information criterion (AIC) defined by $AIC = 2k - 2\log L$, where $k$ is the number of model parameters, and $L$ is the maximized log-likelihood value. The first term of AIC acts as a penalty term to penalize models having many parameters. The second term acts as a measure of fitting to the data with smaller values to be preferred. Therefore, a smaller value of AIC gives a better fit.

Table 1 shows the estimation results. This table summarizes estimates, standard errors calculated by the Hessian matrix of the log-likelihood function [43], and AIC. All estimates are significant at the 5% level on the basis of the Wald test with the null hypothesis that each parameter corresponds to 0. In addition, we can see that the proposed model fits Data-A better than the simple Poisson process model because it leads to a smaller AIC value. According to a heuristic rule (p. 70 of [44]; see also [45]), which is useful for nested models, if a model has more than 10 AIC units lower than another, then it is considered significantly better than that model. In Table 1, as the AIC for the proposed model satisfies the rule (i.e., −1058.03 − (−1267.93)>10), it is a good choice over the Poisson process model.

The estimation result of the proposed model is $\hat{\lambda}_0 = 2.685, \hat{a} = 1.399, \hat{b} = 2.689,$

$\hat{\alpha} = 2.866, \hat{\beta} = 7.626$. Because $\hat{\lambda}_0 = 2.685$, the expected player hours to the first *kyujo*

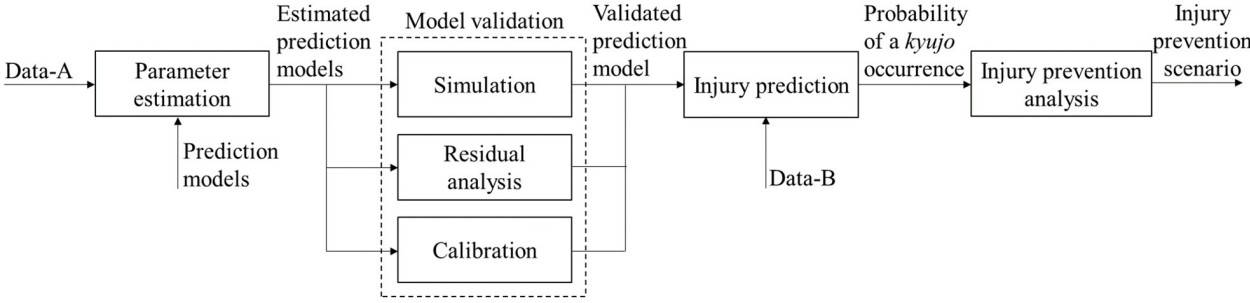

**Fig 5. Overall structure of model validation and injury prediction processes.**

occurrence is $0.372(= 1/\hat{\lambda}_0)$, and $\hat{p}_{i1}(0.015|t) = 0.0395$ (N.B. 0.015 corresponds to the period of a tournament). This result indicates that an average professional sumo wrestler experiences their first *kyujo* occurrence within 0.372 player hours from the first bout in the proposed model. We can also see that each wrestler has a probability of 0.632 for the first *kyujo* occurrence until 0.372 player hours. This is obtained by the following calculation.

$$\hat{p}_{i1}(0.372|t) = 1 - e^{-\hat{\lambda}_0 0.372} = 1 - e^{-1} = 0.632.$$

In other words, this expresses that, on average, each of about 132 wrestlers out of 209 (132 ≈ 0.632 × 209) has a risk of encountering their first *kyujo* occurrence by 372 bouts from their first bouts. After the first *kyujo* occurrence, wrestlers enter Stage II (see Fig 3). In particular, $\hat{b} > 1$ means that wrestlers have a wear-out failure period [34], and $\hat{\alpha} > 0$ indicates that the long-term effect of injuries exists for them.

## Model validation

We validate the estimation result. By simulating [39] the Poisson process and Hawkes process with the estimated intensity function $\hat{\lambda}(t|H_t)$, we obtain the average number of *kyujo* occurrences of a sumo wrestler and its confidence interval [46] with respect to player hours. The 100 $(1 - c)$% confidence interval with a significance level of $c$ is computed by the percentiles of $c/2$ and $1 - c/2$ of simulation results. Fig 6 illustrates the simulation results of the Poisson process model and the proposed model with the historical result given by Data-A. For example, it can be seen that the average number of *kyujo* occurrences of the wrestler is 4.8 for $t = 1.0$ in the proposed model. Fig 6 indicates that the pointwise 99% confidence interval of the proposed model is wider than that of the Poisson process model and covers most of the historical data.

Next, we compare the residuals [30] of these prediction models for a more detailed comparison of the goodness of fit to Data-A. We investigate whether the estimated models can

**Table 1. Estimation results of parameters and AIC for each model.**

| Model | $\hat{\lambda}_0$ | $\hat{a}$ | $\hat{b}$ | $\hat{\alpha}$ | $\hat{\beta}$ | AIC |
|---|---|---|---|---|---|---|
| Poisson process | 4.671* | – | – | – | – | −1058.03 |
| | (0.149) | | | | | |
| Proposed | 2.685* | 1.399* | 2.689* | 2.866* | 7.626* | −1267.93 |
| | (0.170) | (0.298) | (0.300) | (0.630) | (2.663) | |

Standard errors are given in parentheses.

* symbol denotes that a parameter is significant at the 5% level on the basis of the Wald test.

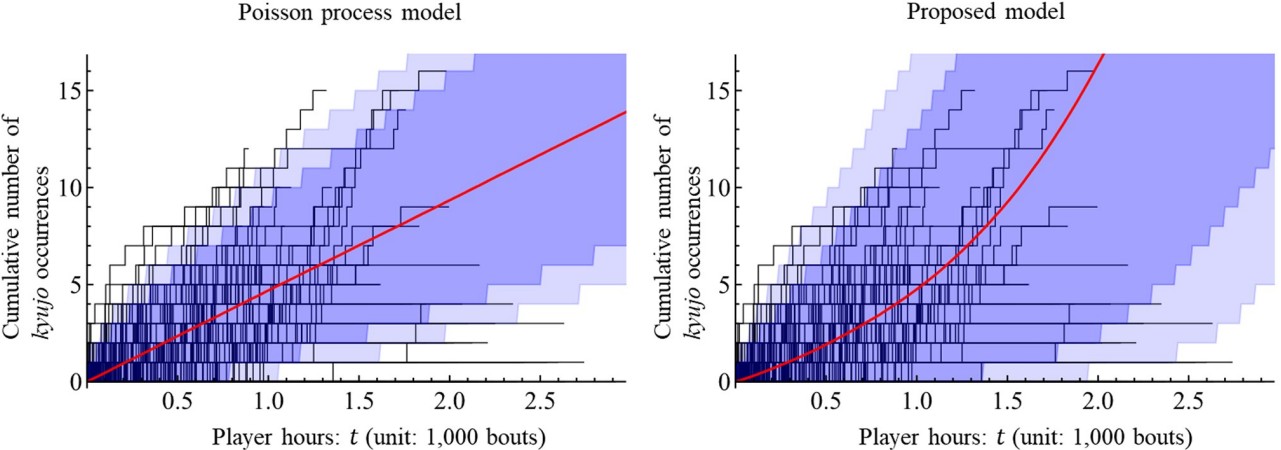

**Fig 6. Behavior of the estimated models on the actual numbers of *kyujo* occurrences of sumo wrestlers.** Each black line represents the actual cumulative numbers of *kyujo* occurrences of 209 wrestlers in Data-A. The red line is the average cumulative number of *kyujo* occurrences of the wrestler by Monte Carlo simulation of 10,000 iterations. The light and dark shaded areas are the pointwise 99% and 95% confidence intervals of the average cumulative number, respectively.

reproduce the major features of the given data. For $i = 1, 2, \ldots, n$, let $\{t_{i1}, t_{i2}, \ldots, t_{iv_i}\}$ be a realization from a point process with a conditional intensity function $\lambda(t|H_t)$. For any consecutive events $t_{ij-1}$ and $t_{ij}$, consider the integral of $\lambda(t|H_t)$ as

$$\Lambda(t_{ij-1}, t_{ij}) = \int_{t_{ij-1}}^{t_{ij}} \lambda(s|H_t)ds.$$

By the random time change, we define $\{\tau_{i1}, \tau_{i2}, \ldots, \tau_{iv_i}\}$ as

$$\tau_{i1} = \int_0^{t_{i1}} \lambda(s|H_t)ds, \qquad \tau_{ij} = \tau_{ij-1} + \Lambda(t_{ij-1}, t_{ij}).$$

Here, it is well known that the duration times $\tau_{ij} - \tau_{ij-1} = \Lambda(t_{ij-1}, t_{ij})$ obey the exponential distribution with mean 1. This sequence data $\{\tau_{i1}, \tau_{i2}, \ldots, \tau_{iv_i}\}$ are called *residuals* [30]. Moreover, $u_{ij} = 1 - \text{Exp}[-\tau_{ij}]$ for $j = 1, 2, \ldots, v_i$ are i.i.d. uniform random variables on [0, 1]. If the estimated intensity $\hat{\lambda}(t|H_t)$ is a good approximation to the true $\lambda(t|H_t)$, the transformed data $\{u_{i1}, u_{i2}, \ldots, u_{iv_i}\}$ are expected to follow the uniform distribution on [0, 1]. Fig 7, which is known as *u-plot* [47], shows the empirical distributions of data $\{u_{i1}, u_{i2}, \ldots, u_{iv_i}\}$ for a sumo wrestler in Data-A transformed by the Poisson process model and proposed model, respectively. The Kolmogorov-Smirnov goodness-of-fit test is a useful way to judge whether the residuals follow the uniform distribution. The residual given by the Poisson process model is not statistically significant at the 5% level, while that given by the proposed model is statistically significant. We performed the same procedure for all 209 wrestlers in Data-A. As a result, residuals of 31 wrestlers are not statistically significant at the 5% level for the Poisson process model, and those of 8 wrestlers are not statistically significant for the proposed model. That is, the proposed model fits Data-A better than the Poisson process model. Therefore, we only consider the proposed model for the injury prediction of wrestlers hereafter.

Finally, we perform calibration [48] of the proposed model to assess its predictive power and goodness of fit to Data-A. Calibration is to check the degree of approximation of the

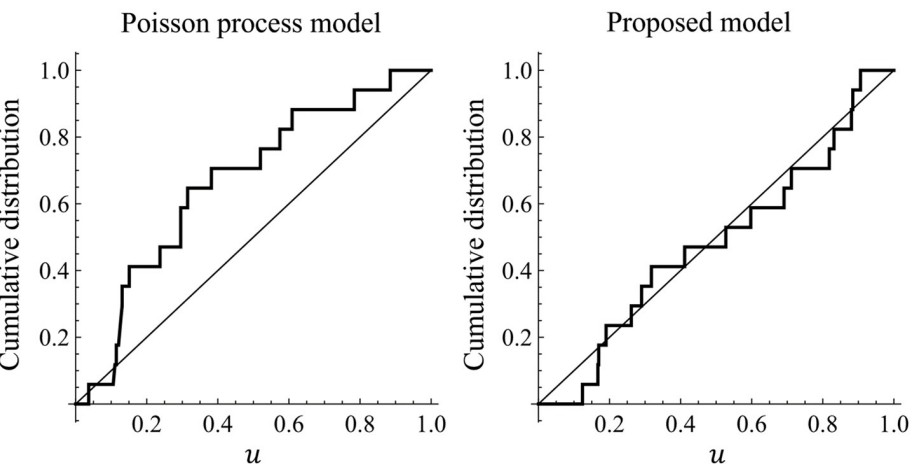

**Fig 7. Empirical distributions of data $\{u_{i1}, u_{i2}, \ldots, u_{iv_i}\}$ for a sumo wrestler in Data-A transformed by the Poisson process model and proposed model, respectively.** Diagonal lines are the cumulative distribution function of the uniform distribution on [0, 1].

predicted probabilities to the actual probabilities. The procedure of the calibration is in the following steps.

1. Calculate the estimated probability of *kyujo* occurrences for all sumo wrestlers in Data-A at each grand sumo tournament from the first bout to retirement. Note that the probability is given by $p_{ij}(0.015|t)$ for the wrestlers in the two top divisions or $p_{ij}(0.007|t)$ for those in the lower-ranked divisions.

2. Divide the range [0,0.3] into a series of intervals with increments of 0.05, and group the estimated probabilities into each interval. Then, calculate the average of the probabilities for each interval as the predicted probability of a *kyujo* occurrence.

3. For each interval, calculate the number of actual *kyujo* occurrences divided by the sample size as the observed probability of a *kyujo* occurrence.

4. Draw a scatter plot with the predicted probability versus the observed one.

Fig 8 illustrates the calibration plot for all wrestlers in Data-A. Sample sizes for the probabilities in the intervals of [0, 0.05], [0.05, 0.1], [0.1, 0.15], [0.15, 0.2], [0.2, 0.25], and [0.25, 0.3] are 9438, 7075, 1485, 217, 39, and 5, respectively. We can see that the proposed model well predicts the actual *kyujo* occurrences on average because the points are plotted on the diagonal line. The point for the probability in the interval of [0.25, 0.3] is farthest away from the diagonal due to the small sample size, which indicates that the proposed model underestimates the predicted probability of a *kyujo* occurrence.

To summarize model validation results, the proposed model fits Data-A better than the Poisson process model and is well-calibrated. Therefore, we use the proposed model as a validated injury prediction model in the next subsection. In addition, these results indicate that the assumption in which a retirement can be regarded as a *kyujo* occurrence does not affect the performance of the injury prediction.

## Illustrative example of injury prediction

Now, we demonstrate injury prediction for sumo wrestlers on the basis of the previous estimation result and model validation. As an illustrative example, we perform injury prediction for

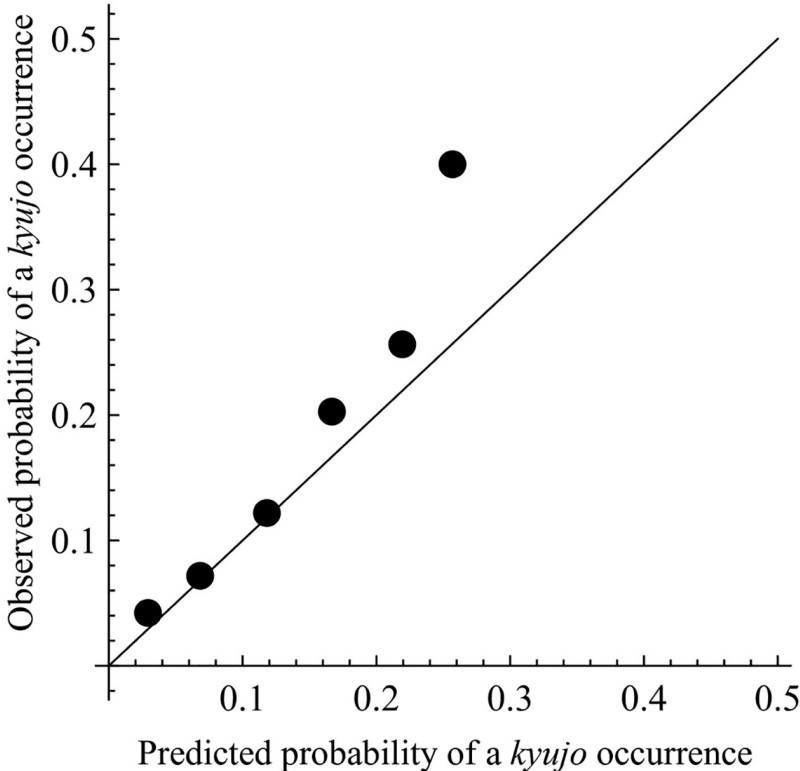

**Fig 8. Calibration plot for the proposed model.**

all wrestlers in Data-B. In the prediction process, we use the proposed model with the estimated parameters. Input data for the model are current player hours $t_i$ and player hours at the $j$-th *kyujo* occurrence of the $i$-th sumo wrestler for $i = 1, 2, \ldots, 42$ and $j = 1, 2, \ldots$, and output data are $\hat{p}_{ij}(0.015|t_i)$, i.e., the estimated probability of the next *kyujo* occurrence for the $i$-th wrestler. Table 2 shows the names and ID numbers of wrestlers included in Data-B. Note that

**Table 2. IDs and names of all sumo wrestlers in the top division of the grand sumo tournament of November 2020.**

| ID | Name | ID | Name | ID | Name |
|----|------|----|------|----|------|
| 1 | Hakuho | 15 | Okinoumi | 29 | Meisei |
| 2 | Kakuryu | 16 | Hokutofuji | 30 | Sadanoumi |
| 3 | Takakeisho | 17 | Tobizaru | 31 | Enho |
| 4 | Asanoyama | 18 | Myogiryu | 32 | Yutakayama |
| 5 | Shodai | 19 | Kotoshoho | 33 | Kaisei |
| 6 | Mitakeumi | 20 | Takarafuji | 34 | Hoshoryu |
| 7 | Takanosho | 21 | Tamawashi | 35 | Ichinojo |
| 8 | Terunofuji | 22 | Tochinoshin | 36 | Chiyonokuni |
| 9 | Takayasu | 23 | Endo | 37 | Kotonowaka |
| 10 | Kiribayama | 24 | Aoiyama | 38 | Chiyotairyu |
| 11 | Wakatakakage | 25 | Terutsuyoshi | 39 | Kotoyuki |
| 12 | Onosho | 26 | Tokushoryu | 40 | Chiyoshoma |
| 13 | Daieisho | 27 | Kotoeko | 41 | Akua |
| 14 | Kagayaki | 28 | Ryuden | 42 | Shimanoumi |

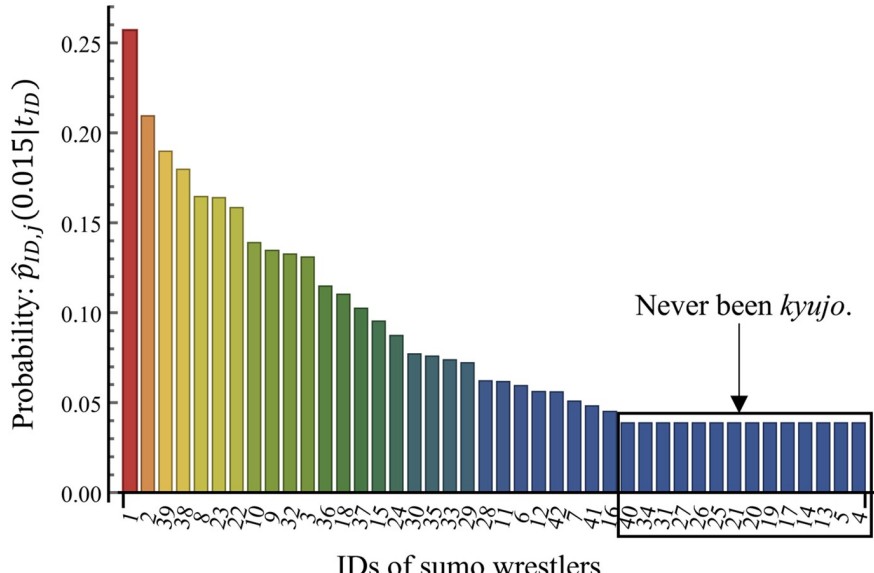

**Fig 9. Estimated probability of *kyujo* occurrences $\hat{p}_{ID,j}(0.015|t_{ID})$ for 42 professional sumo wrestlers who belong to the top division in the grand sumo tournament of November 2020.** ID and names of wrestlers are shown in Table 2.

five wrestlers (ID = 1, 2, 4, 5, 39) actually became *kyujo* due to injuries in the top division of the tournament.

Fig 9 represents the estimated probability of the next *kyujo* occurrence $\hat{p}_{ID,j}(0.015|t_{ID})$ for each sumo wrestler, that is, how likely a wrestler will be injured compared with other wrestlers in the grand sumo tournament. This result contributes to understanding their injury risk. The estimated probability $\hat{p}_{ID,j}(0.015|t_{ID})$ is different for each wrestler. Hakuho (ID = 1 and $j$ = 15) has the highest probability of 25.1% among the 42 wrestlers. Kakuryu (ID = 2 and $j$ = 13) and Tochinoshin (ID = 22 and $j$ = 8) have the second and third highest probabilities of 20.8% and 16.7%, respectively. In addition, 12 wrestlers have the same probability of 3.95% because they have never been *kyujo*. Tamawashi (ID = 21 and $j$ = 1) has the largest player hours (1.286) among these 12 wrestlers. This means that Tamawashi has never been *kyujo* with a probability of $\mathrm{Exp}[-1.286\hat{\lambda}_0] = 0.0337$.

Fig 10 depicts the behavior of the cumulative distribution function $\hat{p}_{ID,j}(w|t_{ID})$ for Hakuho, Kakuryu, and Tochinoshin. From this figure, we can see that the probability of the next *kyujo* occurrence within $w$ player hours for a sumo wrestler having $t_{ID}$ player hours. For example, Hakuho, Kakuryu, and Tochinoshin will be *kyujo* with a probability of more than 50% within 0.05 player hours from the first bout in the grand sumo tournament. Because Hakuho has the largest player hours and number of *kyujo* occurrences among the three, he is most likely to be injured in the future.

We can also estimate the potential number of *kyujo* sumo wrestlers $X$ due to injury in the top division of the grand sumo tournament. Fig 11 shows the estimated distribution of $X$, which summarizes injury risks of wrestlers in the tournament. Such a number is helpful for their coaches to understand an overview of wrestlers' injury risks. The expectation and standard deviation of $X$ are 3.502 and 1.764, respectively. The 95% confidence interval of $X$ is [1, 7], where its lower and upper bounds are computed by the percentiles of 2.5% and 97.5% of $X$,

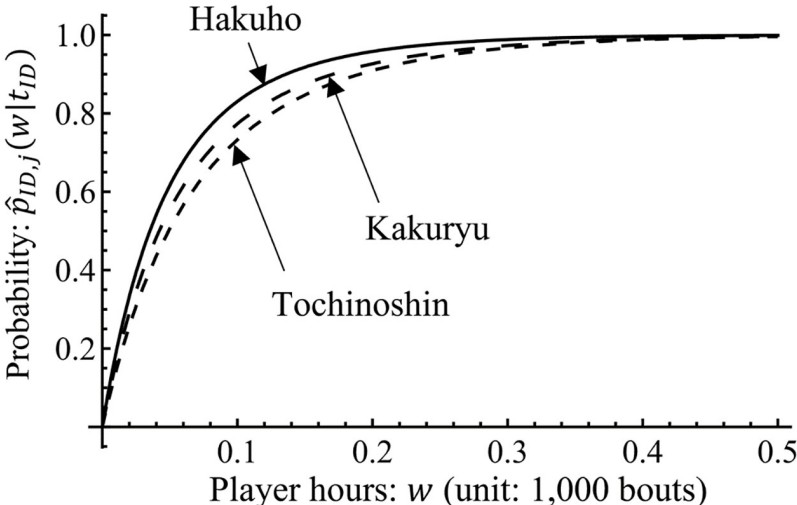

**Fig 10. Probability of the next *kyujo* occurrence within player hours *w* for Hakuho (ID = 1 and *j* = 15), Kakuryu (ID = 2 and *j* = 13), and Tochinoshin (ID = 22 and *j* = 8).** The point of *w* = 0 corresponds to the first bout in the grand sumo tournament of November 2020.

respectively. Because $\hat{q}_5 = 0.138$ is given by Eq (4), we can see that the five wrestlers (ID = 1, 2, 4, 5, 39) actually became *kyujo* with a probability of 13.8% in this tournament.

### Injury prevention scenario

We can use the proposed model to make injury prevention scenarios for professional sumo wrestlers. In general, wrestlers could reduce their injury risk through a long period of *kyujo* for recovery. However, a long period of *kyujo* leads to their rank score being decreased because a *kyujo* is regarded as a loss. From the aforementioned reason, wrestlers tend not to be *kyujo* to maintain their rank score even if they have injuries up to now. Therefore, making a good injury prevention scenario for each wrestler is important so that the wrestler maintains their rank score and prevents potential *kyujo* occurrences in the future.

As Fig 4 shows, the intensity function $\lambda(t|H_t)$ has a decay period after a *kyujo* occurrence, and takes an increasing period after sufficient player hours with no *kyujo* occurrence. This means that the probabilities of *kyujo* occurrences decrease for certain player hours. Therefore, wrestlers should avoid an additional *kyujo* occurrence during the decay period of the intensity function.

For example, let us consider the case of Hakuho (ID = 1). Hakuho had the 10th *kyujo* occurrence at the player hours $t_{1,10} = 1.461$ and additional four *kyujo* occurrences between $t_{1,10}$ and the first day of the grand sumo tournament of November 2020 ($t = 1.582$) in reality. Fig 12 illustrates the behavior of the intensity function of Hakuho in the estimated proposed model. In this figure, the dashed curve means the intensity function under the assumption that Hakuho does not have *kyujo* occurrences after $t_{1,10}$. Now, let us consider the case if Hakuho had consecutive *kyujo* from $t_{1,10}$ to heal his injuries. Note that we assumed that consecutive *kyujo* occurrences were treated as one *kyujo* occurrence in the subsection on statistical modeling.

If Hakuho dared to have consecutive *kyujo* during a grand sumo tournament (i.e., $t \in$ [1.461, 1.476 = 1.461 + 0.015]), the predicted distribution of the number of additional *kyujo* occurrences until $t = 1.582$ is illustrated by Fig 13 through a Monte Carlo simulation using the

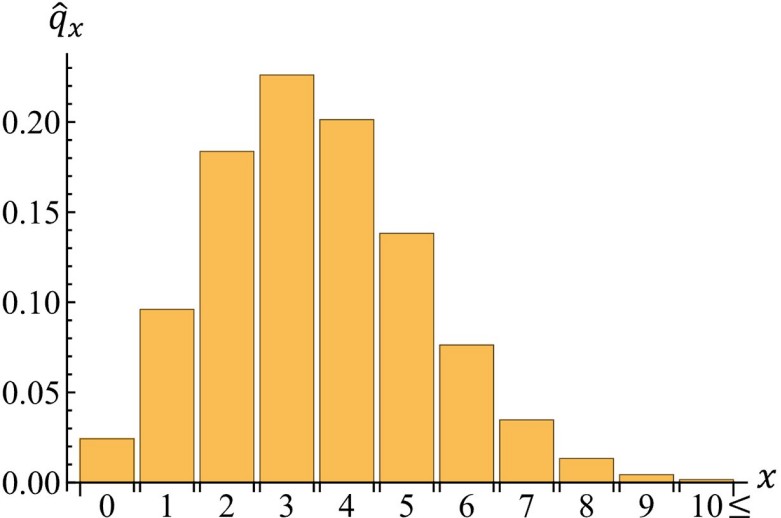

**Fig 11. Estimated distribution of the number of *kyujo* sumo wrestlers in the top division of the grand sumo tournament of November 2020.**

proposed model $\hat{\lambda}(t|H_t)$ with 10,000 iterations. In the simulation result, Hakuho has 1.813 *kyujo* occurrences on average until $t = 1.582$. Thus, Hakuho would avoid $2.227(= 4 - 1.813)$ potential *kyujo* occurrences on average by taking the consecutive *kyujo* after $t_{1,10}$. Note that the probability that Hakuho has four or more *kyujo* occurrences is 0.130 in this simulation. If the reduction of potential *kyujo* occurrences were valuable for Hakuho, he should have planned for the consecutive *kyujo*.

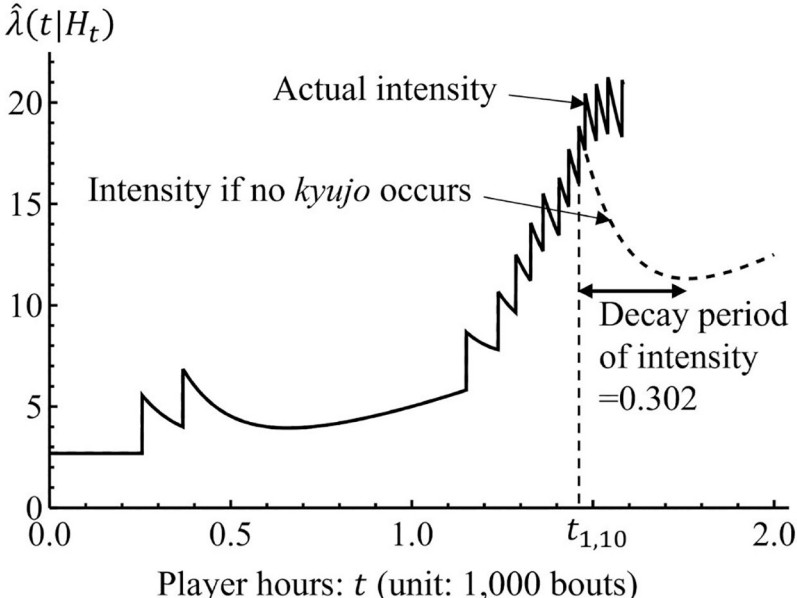

**Fig 12. Behavior of the intensity function of Hakuho (ID = 1) until the grand sumo tournament of November 2020.**

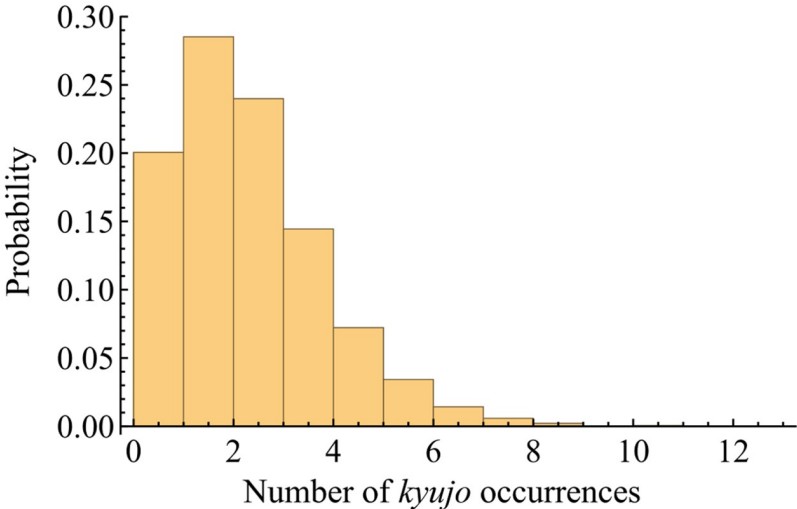

**Fig 13. Hakuho's predicted distribution of the number of *kyujo* occurrences for $t \in [1.461, 1.582]$ through a Monte Carlo simulation using the proposed model $\hat{\lambda}(t|H_t)$ with 10,000 iterations.** N.B. Hakuho had four *kyujo* occurrences for $t \in [1.461, 1.582]$ in reality.

In the same way, we can make injury prevention scenarios for professional sumo wrestlers quantitatively by simulating the proposed model. For example, wrestlers can make a decision whether it is better to be *kyujo* than to participate in grand sumo tournaments when they have injuries. In addition, they can plan appropriate *kyujo* periods if they should be *kyujo*. Such a means of risk-based scenario planning is novel and beneficial for wrestlers and their coaches. In future work, further study is expected from the viewpoint of operations research to optimize decision-making for injury prevention.

## Conclusion

In this paper, we have investigated injury prediction for professional sumo wrestlers. Through the feature extraction of actual data, we determined the characteristics of *kyujo* occurrences and have proposed a statistical model of *kyujo* occurrences. As a result of the parameter estimation of $\hat{\alpha} > 0$, we have also found the long-term effect of *kyujo* occurrences. The proposed model provides the estimated probability of an injury occurrence for a wrestler and the predicted number of *kyujo* wrestlers due to injury in a grand sumo tournament. The estimated probabilities help communicate the risk of injury to wrestlers.

The proposed model can be also applied to injury prediction for athletes of other sports if the failure rate of the athletes behaves similarly to that of sumo wrestlers. One can estimate the model parameters by only historical count data of injury occurrences. Some modification of the proposed model will be necessary to consider characteristics that are uniquely observed for injury occurrences of athletes. For example, one should add trend terms to Eq (2) if there is a seasonality in injury occurrences. After model validation, injury prediction for athletes is applicable by computing $p_{ij}(w|t)$ given $w$ and $t$.

A limitation of the proposed model based on Data-A is that it would underestimate the risk of relatively minor injuries. Because we only used player hours of prior *kyujo* occurrences as the input, the proposed model based on Data-A cannot predict acute, short, and moderate injuries. To solve the limitation, we need both *kyujo* and injury data from daily observations. This should enable us to modify the proposed model by using the new data.

Our ongoing work focuses on studying the effect of individual characteristics, especially modifiable risk factors, such as age, height, and weight on injury risk. By extending the proposed model to consider the characteristics, we can evaluate the relationship between injury occurrences and the characteristics. For example, we can determine how the probabilities of injury occurrences increase/decrease by using the extended model if athletes gain/lose weight. To extend the model, it is expected to consider the concept of survival analysis. In this paper, we modeled *kyujo* occurrences by the Hawkes process to consider the behavior of intensity changes. This model structure is also handled in survival analysis under the guise of multiple events, although typically without assuming intensity changes as the Hawkes process does. However, one could also implement a time-varying covariate that captured the number of previous *kyujo* occurrences by using multistate models [49, 50].

## Supporting information

**S1 File. Code of parameter estimation and simulation for the models.**
(NB)

## Author Contributions

**Conceptualization:** Shuhei Ota.

**Data curation:** Shuhei Ota.

**Formal analysis:** Shuhei Ota, Mitsuhiro Kimura.

**Funding acquisition:** Shuhei Ota, Mitsuhiro Kimura.

**Investigation:** Shuhei Ota, Mitsuhiro Kimura.

**Methodology:** Shuhei Ota, Mitsuhiro Kimura.

**Project administration:** Shuhei Ota.

**Resources:** Shuhei Ota.

**Software:** Shuhei Ota.

**Supervision:** Mitsuhiro Kimura.

**Validation:** Shuhei Ota, Mitsuhiro Kimura.

**Visualization:** Shuhei Ota, Mitsuhiro Kimura.

**Writing – original draft:** Shuhei Ota, Mitsuhiro Kimura.

**Writing – review & editing:** Shuhei Ota, Mitsuhiro Kimura.

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
