## [Decision Letter · Decision Letter 0]

19 Dec 2022

PONE-D-22-28529Statistical injury prediction for professional sumo wrestlers: modeling and perspectivesPLOS ONE

Dear Dr. Ota,

Thank you for submitting your manuscript to PLOS ONE. After careful consideration, we feel that it has merit but does not fully meet PLOS ONE’s publication criteria as it currently stands. Therefore, we invite you to submit a revised version of the manuscript that addresses the points raised during the review process.

We look forward to receiving your revised manuscript.

Kind regards,

Viacheslav Kovtun, Dr.Sc., Ph.D.

Academic Editor

PLOS ONE

Journal Requirements:

2. PLOS requires an ORCID iD for the corresponding author in Editorial Manager on papers submitted after December 6th, 2016. Please ensure that you have an ORCID iD and that it is validated in Editorial Manager. To do this, go to ‘Update my Information’ (in the upper left-hand corner of the main menu), and click on the Fetch/Validate link next to the ORCID field. This will take you to the ORCID site and allow you to create a new iD or authenticate a pre-existing iD in Editorial Manager. Please see the following video for instructions on linking an ORCID iD to your Editorial Manager account: https://www.youtube.com/watch?v=_xcclfuvtxQ.

Reviewers' comments:

Reviewer's Responses to Questions

**Comments to the Author**

1. Is the manuscript technically sound, and do the data support the conclusions?

Reviewer #1: Partly

Reviewer #2: Yes

2. Has the statistical analysis been performed appropriately and rigorously? 

Reviewer #1: No

Reviewer #2: Yes

3. Have the authors made all data underlying the findings in their manuscript fully available?

Reviewer #1: Yes

Reviewer #2: Yes

4. Is the manuscript presented in an intelligible fashion and written in standard English?

Reviewer #1: Yes

Reviewer #2: Yes

5. Review Comments to the Author

Reviewer #1: Review Report On

Paper ID: PONE-D-22-28529

Paper Title: Statistical injury prediction for professional sumo wrestlers: modeling and perspectives

The study proposes a model for predicting the injury occurrences for sumo wrestlers. The new model is a combination of the existing Poisson and the Binomial model that is based in the Hawkes process. The authors are reporting that the model is superior to the existing one in terms of the model selection criteria. Some comments regarding this study are as below:

(1) Major comments

1. What are the necessary assumption(s) of the proposed model? Are those same as the Poisson model?

2. If a model is more than 2 AIC units lower than another, then it is considered significantly better than that model. In Table 1, the AIC for the proposed model is not so and hence may not be the good choice over the Poisson model.

3. Generally, a large confidence interval indicates the sample does not provide a precise representation of the population parameter estimate, whereas a narrow confidence interval demonstrates a greater degree of precision. The model validation part of this paper is saying that large/wider confidence interval is obtained for the proposed model and the model is good one. This totally opposite of the general existing process.

4. Also, the prediction process is not well defined.

5. Why validation is done only at 5% and 10% level of significance? Why not for the 1%?

(2) Minor comments

1. The term operations research is mentioned in “Keywords”—what does this concept mean here?

Reviewer #2: The authors present an interesting application of the Hawkes process in modeling the occurrence of kyujo across sumo players' careers.

Since the authors use maximum likelihood estimation, it should be possible to provide basic results for asymptotic normality such as standard errors and Wald-based tests. See, e.g., Serfling (2009) for basic theoretical discussion. Most statistical packages that implement nonlinear estimation should also provide these outputs.

The kyujo data are subject to censoring. It is unclear what effect this censoring has on the estimation of the underlying failure model. The authors should investigate this to at least some degree. It seems reasonable that with multiple events per player that the effects might be minimal. Is it possible to account for the censoring process in the model estimation?

The kyujo data appear to be at least partially subject to informative censoring. That is, while players most likely end their careers for a variety of reasons, it seems that the injuries leading to kyujo or repeated kyujo are likely the proximal cause in many cases. In the survival analysis field, this type of informative censoring can sometimes be dealt with via the method of inverse probability of censoring weights (IPCW).

One relatively simple way to handle this would be to set up simulation studies that implement more or less informative censoring. For estimation performance, it would be of interest anyway to see how well the model performs in retrieving model parameters in simulation study.

The authors write (Line 247) that "the proposed model fits Data-A better than the Poisson process model". This is not really immediately evident from the inspection, given the noise in the data toward the tail end of Figure 4. Is there any way of defining residuals for this model, perhaps similarly to those based on counting processes?

If the authors can devise a calibration plot similar to those found in logistic regression, it might be useful not only in displaying the predictive power but also in evaluating goodness of fit.

In the Data-B model fit, the presentation of the predicted number versus the actual number of kyujo is somewhat lost in the text. The text from Lines 276-282 is somewhat opaque. What is being said here? It seems like some of the players with higher predicted risk did in fact become kyujo, but it requires additional work by the reader to figure it out. It is not clear that a confidence interval for number of kyujo is highly informative, but one could construct one from the predicted distribution.

This general model structure is also handled in survival analysis under the guise of multiple events, although typically without assuming intensity changes as the Hawkes process does. However, one could also implement a time-varying covariate that captured the number of previous kyujo as an approximation. See, e.g., Therneau and Grambsch (2000) for discussion of this model as well as multi-state models which could also be of interest. Could the authors comment briefly on this?

Line 200: Change to "subsection".

References:

Serfling, R. J. (2009). Approximation theorems of mathematical statistics. John Wiley & Sons.

Therneau, T.M. and Grambsch, P.M. (2000) Modeling Survival Data: Extending the Cox Model. Springer Science & Business Media.

6. PLOS authors have the option to publish the peer review history of their article (what does this mean?). If published, this will include your full peer review and any attached files.

Reviewer #1: No

Reviewer #2: No

---

## [Author Response · Author response to Decision Letter 0]

17 Feb 2023

Please see the attached file of the response letter.

---

## [Decision Letter · Decision Letter 1]

6 Mar 2023

Statistical injury prediction for professional sumo wrestlers: modeling and perspectives

PONE-D-22-28529R1

Dear Dr. Ota,

We’re pleased to inform you that your manuscript has been judged scientifically suitable for publication and will be formally accepted for publication once it meets all outstanding technical requirements.

Kind regards,

Viacheslav Kovtun, Dr.Sc., Ph.D.

Academic Editor

PLOS ONE

Additional Editor Comments (optional):

Reviewers' comments:

Reviewer's Responses to Questions

**Comments to the Author**

1. If the authors have adequately addressed your comments raised in a previous round of review and you feel that this manuscript is now acceptable for publication, you may indicate that here to bypass the “Comments to the Author” section, enter your conflict of interest statement in the “Confidential to Editor” section, and submit your "Accept" recommendation.

Reviewer #1: All comments have been addressed

Reviewer #2: All comments have been addressed

2. Is the manuscript technically sound, and do the data support the conclusions?

Reviewer #1: Partly

Reviewer #2: (No Response)

3. Has the statistical analysis been performed appropriately and rigorously? 

Reviewer #1: Yes

Reviewer #2: (No Response)

4. Have the authors made all data underlying the findings in their manuscript fully available?

Reviewer #1: Yes

Reviewer #2: (No Response)

5. Is the manuscript presented in an intelligible fashion and written in standard English?

Reviewer #1: Yes

Reviewer #2: (No Response)

6. Review Comments to the Author

Reviewer #1: (No Response)

Reviewer #2: (No Response)

7. PLOS authors have the option to publish the peer review history of their article (what does this mean?). If published, this will include your full peer review and any attached files.

Reviewer #1: No

Reviewer #2: No

---

## [Editor Report · Acceptance letter]

9 Mar 2023

PONE-D-22-28529R1 

Statistical injury prediction for professional sumo wrestlers: modeling and perspectives 

Dear Dr. Ota:

I'm pleased to inform you that your manuscript has been deemed suitable for publication in PLOS ONE. Congratulations! Your manuscript is now with our production department. 

Kind regards, 

on behalf of

Professor Viacheslav Kovtun 

Academic Editor

PLOS ONE